# Efficient Propylene Carbonate Synthesis from Urea and Propylene Glycol over Calcium Oxide–Magnesium Oxide Catalysts

**DOI:** 10.3390/ma16020735

**Published:** 2023-01-11

**Authors:** Kavisha Dang, Navneet Kumar, Vimal Chandra Srivastava, Jinsub Park, Mu. Naushad

**Affiliations:** 1Department of Chemical Engineering, Indian Institute of Technology Roorkee, Roorkee 247667, India; 2Department of Electronics Engineering, Hanyang University, Seoul 04763, Republic of Korea; 3Department of Chemistry, College of Science, King Saud University, P.O. Box 2455, Riyadh 11451, Saudi Arabia

**Keywords:** cyclic carbonates, alcoholysis, propylene carbonate, propane-1,2-diol, urea

## Abstract

**Highlights:**

**Abstract:**

A series of calcium oxide–magnesium oxide (CaO–MgO) catalysts were prepared under the effects of different precipitating agents and using varied Mg/Ca ratios. The physiochemical characteristics of the prepared catalysts were analyzed using XRD, FE-SEM, BET, FTIR, and TG/DTA techniques. Quantification of basic active sites present on the surface of the CaO–MgO catalysts was carried out using the Hammett indicator method. The as-prepared mixed oxide samples were tested for propylene carbonate (PC) synthesis through the alcoholysis of urea with propylene glycol (PG). The effects of the catalyst composition, catalyst dose, reaction temperature, and contact time on the PC yield and selectivity were investigated. The maximum PC yield of 96%, with high PC selectivity of 99% and a urea conversion rate of 96%, was attained at 160 °C using CaO–MgO catalysts prepared using a Mg/Ca ratio of 1 and Na_2_CO_3_ as a precipitating agent. The best-performing catalysts also exhibited good reusability without any significant loss in PC selectivity. It is expected that the present study will provide useful information on the suitability of different precipitating agents with respect to the catalytic properties of the oxides of Ca and Mg and their application in the synthesis of organic carbonates.

## 1. Introduction

Organic carbonates constitute an important class of green solvents due to their low eco-toxicity and their biodegradability. They have been widely employed as solvents in paints, cosmetics, polymer coatings, food packaging, and adhesives [1,2,3]. They are also employed as solvents for organic transformations due to their high stability under ambient conditions [4]. Organic carbonates can be categorized as linear organic carbonates (such as dimethyl carbonate and diethyl carbonate) or cyclic organic carbonates (such as ethylene carbonate, propylene carbonate, and glycerol carbonate) [5,6,7]. Among these, propylene carbonate (PC) can also be used as a starting material for preparing other organic carbonates, such as dimethyl carbonate, diethyl carbonate, and glycerol carbonate [8,9,10,11]. Additionally, it has found application as an electrolyte for Li-ion batteries due to its high dielectric constant [12,13,14]. Therefore, its increased consumption has led to demand for its preparation in large amounts. Continuous efforts have been made to develop potential catalysts to synthesize propylene carbonates in high amounts.

The oldest method used for the synthesis of PC is phosgenation [15]. This involves the reaction of propylene glycol and phosgene, with the production of HCl as a byproduct. The use of toxic raw materials (i.e., phosgene) is the major drawback of this method. Another approach for PC synthesis is the oxidative carbonylation of alcohols with carbon monoxide. Despite showing good product selectivity, this route also suffers from the drawback of using a toxic and poisonous gas as a raw material. The carbonate transesterification route leads to the formation of azeotropes, making the separation of products difficult [4]. PC synthesis by carbonylation of alcohols is an environmentally friendly method, as it involves the utilization of unwanted CO_2_ gas [16,17,18]. However, the thermodynamic and kinetic limitations of this route lead to low conversion and low yield of PC.

PC synthesis from urea alcoholysis with PG is a popular and beneficial method that uses raw materials that are cheap and easily available and proceeds under mild reaction conditions. No azeotrope formation takes place, enabling the easy separation of the product from the reaction mixture. NH_3_ that is produced as a byproduct can be recycled, leading to improved product yield [19].

A number of homogeneous and heterogeneous catalysts have been used to catalyze the reaction of urea alcoholysis for PC synthesis [20,21,22,23]. Homogeneous catalysts are completely miscible in the reactant mixture, providing a better degree of interaction between the reactant molecules and the catalyst [24]. On the other hand, heterogeneous catalysts can be easily separated from the product, regenerated, and reused, making the overall process more economical. Despite this advantage, heterogeneous catalysts face difficulties in reusability due to the deformation of their structure [3]. The requirement of a high reaction temperature is another common barrier. Some synthesis routes are hazardous to health, involve the use of poisonous raw materials, and cause pollution at the same time, which constitutes a significant limitation. These shortcomings have spurred the need to explore new catalysts.

A number of studies have been devoted to the usage of transition and alkaline earth metal oxides for the selective production of PC via the process of urea alcoholysis. For example, a high PC yield of 94.8% in 30 min was recorded using Zn–Mg mixed oxides due to the synergistic effect between Mg and Zn and improved surface basicity [25]. In another study, a ternary oxide of Zn–Ca–Al exhibited high PC selectivity (98.4%) and yield (90.8%) because of the increased basicity resulting from the deposition of CaO And Ca(OH)_2_ species on the catalyst’s surface [26]. Several other studies also demonstrated improved PC production over catalysts composed of Mg and Ca, due to the alteration in the number of basic sites [27,28]. Overall, it can be inferred that Ca and Mg furnish basic active sites to the catalysts and can promote urea alcoholysis to produce PC. Moreover, these elements are low-cost and highly abundant in nature. Therefore, it is worth testing the performance of calcium oxide–magnesium oxide catalysts for PC production using urea and PG as starting materials. To the best of the authors’ knowledge, no work has been reported on the utilization of CaO–MgO for PC synthesis through the urea alcoholysis route. In the present study, CaO–MgO catalysts were prepared with different Mg/Ca molar ratios. The effects of different precipitating agents (e.g., NaOH, Na_2_CO_3_) and a mixture of NaOH/Na_2_CO_3_ on the physical and chemical properties of the catalyst were also investigated. Furthermore, the PC synthesis experiments were performed under varied reaction parameters (i.e., reaction temperature, reaction time, PG/urea molar ratio, and catalyst dosage) to investigate their effects on PC yield and selectivity. Reusability studies on the spent catalysts were performed to verify the industrial viability of CaO–MgO catalysts.

## 2. Materials and Methods

### 2.1. Materials and Reagents

All chemicals used for the experiment were of analytical grade. The main raw materials used for the reaction were propylene glycol of 99.5% purity (AR, Loba Chemie Pvt. Limited, Mumbai, India) and urea (extra pure, SD Fine Chemicals Limited, Mumbai, India). The chemicals used for the synthesis of the catalysts were calcium nitrate tetrahydrate, Ca(NO_3_)_2_.4H_2_O (AR, Hi-Media Laboratories, India); magnesium nitrate hexahydrate, Mg(NO_3_)_2_.6H_2_O (AR, SD Fine Chemicals Limited, Mumbai, India); sodium hydroxide, NaOH (AR, Fischer Scientific Limited, Mumbai, India); and sodium carbonate anhydrous, Na_2_CO_3_ (AR, Hi-Media Laboratories, Mumbai, India).

### 2.2. Catalyst Preparation

CaO–MgO catalysts were synthesized using the co-precipitation method by parallel-flow precipitation. Mixtures of Ca(NO_3_)_2_.4H_2_O and Mg(NO_3_)_2_.6H_2_O at different Mg/Ca molar ratios of 0.5, 1, and 2 were dissolved in 100 mL of Millipore water and then stirred magnetically to obtain the cation solution. The anion solution in Millipore water was prepared using different precipitating agents, namely, NaOH, Na_2_CO_3_, and a mixture of NaOH/Na_2_CO_3_. To a beaker containing 50 mL of Millipore water, both of the solutions were added drop by drop. To ensure proper mixing, the solution in the beaker was stirred at a rate of 450 rpm. By adding the requisite volume of the anion solution, the overall pH was maintained at 10.5. The precipitate obtained in this manner was kept for aging at 80 °C for 24 h. This was followed by centrifuging at 8000 rpm—first with Millipore water, and then with ethanol—to remove the dissolved and ionic impurities. It was then neutralized by washing with Millipore water and dried at 100 °C for 16 h. The dried solid was then crushed in a mortar using a pestle to obtain a fine catalyst powder. The last step consisted of calcining the obtained powder at 700 °C for 5 h. The as-prepared catalysts were denoted as MgCaXCNa, MgCaXCN, and MgCaXCNc, where X = Mg/Ca molar ratio (0.1, 1, or 2) and CNa, CN, and CNc indicate co-precipitation (C) of catalysts in the presence of NaOH, Na_2_CO_3_, and a mixture of NaOH/Na_2_CO_3_, respectively.

### 2.3. Catalyst Characterization

TGA analysis of the uncalcined catalyst was performed using an SII 6300 EXSTAR analyzer. The sample was heated from a temperature of 30 °C to a maximum temperature of 1000 °C in an air atmosphere. The heating rate was kept constant at 10 °C/min. X-ray crystallography was used to obtain information about the atomic and molecular structures of the catalyst. A Bruker D8 ADVANCE diffractometer was used. A copper target was used for the generation of K_α_ radiation with a wavelength of 0.154 nm, used for detection of the spacing between the lattice planes of the crystal. X-ray radiation was generated by an X-ray tube (operating at 40 kV and 30 mA). The incident angle θ was continuously changed to obtain the diffractogram of diffraction intensity versus the diffraction angle 2θ (2θ varying from 5–90°, with a step size of 0.02°, at a scanning speed of 2°/min). The surface properties and composition of the catalyst were determined with the help of the FE-SEM model Quanta 200 FEG. EDX results were also generated. The testing was carried out at a voltage of 20 kV. The images were captured at a resolution varying from 2500 to 50,000. The internal surface area was assessment by the adsorption and condensation of N_2_ (using a Micromeritics ASAP 2060 instrument). FTIR spectroscopy was used to collect spectral data over a wide wavelength range of 4000–450 cm^−1^. The obtained spectrograph was used to gather information about the functional groups present on the surface of the catalyst. A Thermo Nicolet Magna 760 FTIR spectrophotometer was used for all of the analyses.

The Hammett indicator method was used to determine the basicity of the catalysts prepared using varied molar ratios of Mg/Ca and different precipitating agents [29]. Briefly, an indicator solution of bromothymol blue was prepared by adding 12.8 g of bromothymol blue to 10 mL of benzene. A 0.1 N benzoic acid solution in benzene was used as a titrant. A fixed mass of various catalysts (i.e., 0.5 g) was suspended in 20 mL of benzene in a titrating flask, followed by the addition of 1 mL of the indicator solution to yield a green-colored titrate. The titration analysis was performed until the greenish color of the solution disappeared to reflect the presence of basic sites on the catalyst surface [19]. The volume of the titrant required to decolorize the solid was noted. The basicity was calculated in mmol/g using the following equation; note that the analysis was performed in triplicate, and the results with error are presented in Table 1:(1)$$\mathrm{Basicity}\;\left(\frac{\mathrm{mmol}}{g}\right)=\frac{\mathrm{Molarity}\;\mathrm{of}\;\mathrm{the}\;\mathrm{titrant}\;\mathrm{solution}\;\times \;\mathrm{Volume}\;\mathrm{of}\;\mathrm{the}\;\mathrm{titrant}\;\mathrm{required}}{\mathrm{Weight}\;\mathrm{of}\;\mathrm{the}\;\mathrm{catalyst}\;\mathrm{sample}\;\mathrm{taken}\;\mathrm{initially}}$$

### 2.4. PC Synthesis from Urea and PG

An autoclave reactor with 50 mL capacity was used for carrying out the synthesis reaction. In a typical reaction, 30 mL of PG, 6.15 g of urea (PG-to-urea molar ratio of 4:1), and 0.4 g of catalyst (1 wt.% PG and urea combined weight) were added to the reaction vessel and then clamped into the reactor. The reaction temperature was set to 160 °C, and a vacuum pump was used to remove the ammonia gas produced during the reaction. The reaction was carried out for 3 h. After the reaction, the product formed was taken out and allowed to cool. The next step involved centrifugation of the product mixture to recover the catalyst particles. The volume of the segregated liquid product was measured and used for further analysis. Note that all experiments were performed in triplicate, and the results are presented with error bars. The error being recorded for PC production under variable experimental parameters was found be less than 5%, which is below the acceptable amount.

Gas chromatography was performed on the reaction product to analyze the products formed. A Michro-9100 chromatograph equipped with an HP-5 capillary column of 50 m × 0.32 mm × 0.17 μm ((5%-phenyl)-methyl poly-siloxane) was used for the analysis. FID was used for the detection of the hydrocarbon compounds formed. The sample for analysis was prepared by adding 1 mL of standard (iso-propyl alcohol (IPA)) to 1 mL of the reaction product obtained after centrifugation and segregation. The initial temperature of the column was kept at 80 °C and was raised to 270 °C at a rate of 15 °C/min. The column inlet and detector temperature were kept at 280 °C. The selectivity of PC, conversion of urea, and yield of PC were calculated by the following equations:(2)$$\mathrm{Selectivity}\;\mathrm{of}\;\mathrm{PC}\;(\%)=\frac{\mathrm{Moles}\;\mathrm{of}\;\mathrm{PC}\;\mathrm{formed}}{\mathrm{Total}\;\mathrm{moles}\;\mathrm{of}\;\mathrm{product}\;\mathrm{formed}}\times 100$$
(3)$$\mathrm{Conversion}\;\mathrm{of}\;\mathrm{urea}\;(\%)=\frac{\mathrm{Total}\;\mathrm{moles}\;\mathrm{of}\;\mathrm{products}\;\mathrm{formed}}{\mathrm{Initial}\;\mathrm{mole}\;\mathrm{of}\;\mathrm{urea}\;\mathrm{taken}}\times 100$$
(4)$$\mathrm{Yield}\;\mathrm{of}\;\mathrm{PC}\;(\%)=\frac{\mathrm{Moles}\;\mathrm{of}\;\mathrm{PC}\;\mathrm{formed}}{\mathrm{Initial}\;\mathrm{moles}\;\mathrm{of}\;\mathrm{urea}\;\mathrm{taken}}\times 100$$

## 3. Results and Discussion

### 3.1. Catalyst Characterization

The thermal analysis of the uncalcined Ca–Mg precursors prepared with three different precipitants is shown in Figure 1. A significant reduction in the weight of all of the catalysts was observed at two temperatures. An approximately 17% loss in weight was observed below 400 °C, attributed to the transformation of precipitated Mg compounds into MgO and the evaporation of solvent water and the water of crystallization [30]. Extensive weight reduction was observed in the temperature range of 400–700 °C. This was observed because of the decomposition of carbonates. Oxidation and combustion of carboxyl and nitrate groups might be another possible reaction responsible for the emission of CO_2_, N_2_, and water vapor. The final reduction of the precursor occurred around 700 °C. Hence, 700 °C was chosen as the calcination temperature for the production of CaO–MgO [31].

Powdered XRD diffraction peaks for CaO–MgO synthesized with varying Mg/Ca ratios and types of precipitants are shown below in Figure 2. The XRD peaks of the catalysts showed the characteristic peaks of distinct phases of CaO and MgO, without the presence of any mixed-oxide phase of Ca and Mg oxide particles. The large difference in the ionic radii of the Ca^2+^ and Mg^2+^ ions was the reason for the formation of a composite oxide instead of a mixed oxide [32]. This presence of cubic CaO (2θ = 32.2°, 37.3°, and 53.8°) (JCPDS File No. 37-1497) and hexagonal MgO (2θ = 42.9° and 62.3°) (JCPDS File No. 4-0829) was indicated by the diffraction peaks. The diffraction patterns were compliant with the standards of the JCPDS. Upon increasing the Mg/Ca molar ratio, the diffraction peaks of MgO became more intense, while those of CaO decreased, verifying the presence of pure-oxide crystalline phases of Ca and Mg. The catalyst contained the mineral calcite (CaCO_3_) at 2θ of 29.48° and 50.5° [20]. This shows that the CaCO_3_ did not completely decompose to CaO upon calcination. This is because CaCO_3_ has a stable structure with a very hard shell layer that is difficult to decompose at a temperature of 700 °C [33]. On the other hand, some peaks of Ca(OH)_2_ also appeared at 2θ of 17.8°, 28.6°, and 46.8° [34]. The formation of Ca(OH)_2_ was due to the reaction between CaO and H_2_O in air.

The XRD diffractogram for the catalyst prepared using Na_2_CO_3_ showed the sharpest and most intense peaks of CaO and MgO, revealing that the sample was in a highly crystalline state. The absence of peaks at 50.5° corresponding to calcium carbonate as calcite [35] and the peaks at 28.6° and 46.8° corresponding to calcium hydroxide were confirmed by the FTIR and TGA analyses for the catalyst prepared using NaOH.

A comparison of the catalysts’ surfaces before and after calcination is presented in Figure 3. This helped in studying the alterations induced in the structure of CaO–MgO as a result of calcination. It was observed that the layers present before calcination were broken to produce separate CaO and MgO. The calcination at 700 °C transformed the aggregate flakes into smaller spheres, possibly due to the extensive loss of hydroxide and carbonate groups and the shrinkage of the unit cell.

SEM images showed the morphology of the CaO–MgO catalysts with non-uniform structures. Particles of two evidently different types were observed in the images. The tiny particles were of MgO, while the large aggregate particles were of CaO. The catalyst sample with compact agglomerations was consistent with the previously noted morphology of dense particles with a heterogeneous distribution of particle sizes for MgO/CaO [26].

The BET results of the catalysts are displayed in Table 1. It was concluded that the surface area of the catalysts increased when increasing the Mg/Ca ratio from 0.5 to 1. However, upon further increasing the ratio to 2, a decline in the surface area was observed. The surface area of the catalysts prepared with Mg/Ca ratios of 0.5 and 2 was high compared to the catalysts prepared with a Mg/Ca ratio of 1. A possible reason for this decrease in the area could be the aggregation of crystals. Thus, the Mg/Ca ratio of 1 was found to provide the maximum surface area [25].

Figure 4a illustrates the FTIR transmittance spectrum of the CaO–MgO catalyst. Broad peaks at ~3438 cm^−1^ and 3690 cm^−1^ correspond to OH stretching vibrations assigned to the water of crystallization and the interlayer water. The peak at 1460 cm^−1^ corresponds to the asymmetric stretching vibration modes of the carbonate group CO_3_^2−^ [36]. The bands at ~801 and 862 cm^−1^ correspond to the CO_3_^2–^ bending vibration modes of CaCO_3_ [37]. After calcination of the catalysts, the characteristic absorption band of C=O was observed between 2000 and 1500 cm^−1^, indicating the presence of calcium carbonate over the calcined catalyst MgCa_0.5_CNc. The broad band at ~1463 cm^−1^ and the sharp band at 3634 cm^−1^ are associated with the OH stretching vibration mode of water physisorbed on the surface of CaO. This is related to the OH in Ca(OH)_2_, as shown in the XRD pattern. However, these peaks corresponding to water and carbonate were found to be less intensified due to the slight removal of the components in the catalyst prepared with Na_2_CO_3_ as a precipitant. Furthermore, bands of CaO appeared at around 580 cm^−1^. Meanwhile, these peaks almost disappeared in the case of the calcined catalyst prepared with NaOH as a precipitant. This can also be confirmed from the XRD results of MgCa_0.5_CNa, showing weak peaks for Ca(OH)_2_, CaCO_3_, and CaO. The peaks at 705 cm^−1^ were due to the MgO band. The basicity of the various catalysts, as estimated by the Hammett indicator method, is displayed in Figure 4b and summarized in Table 1. As can be seen, various catalysts revealed varying levels of basicity due to differences in the amounts of their constituent components (i.e., CaO, MgO, CaCO_3_, and Ca(OH)_2_) and the nature of the precipitating agents. Among the various catalysts, the strongest basic character (24.07 mmol/g) was demonstrated by MgCa1CN, owing to the existence of various components in desirable amounts.

### 3.2. Effects of Preparation Conditions of CaO–MgO on the Catalytic Performance

#### 3.2.1. Effects of Different Precipitating Agents

Na_2_CO_3_, NaOH, and NaOH + Na_2_CO_3_ were separately used as precipitating agents for the preparation of the catalysts. The best conversion and yield for PC were obtained by the catalysts prepared from Na_2_CO_3_. The highest PC yield obtained was 96%, as can be seen in Figure 4c. This result can also be confirmed from the XRD diffractogram, which shows the sharpest peak for the catalyst prepared by Na_2_CO_3_. This might be because Na_2_CO_3_ was the most effective precipitating agent for causing the precipitation of the salts. This can be explained by the number of active basic sites that were available to catalyze the reaction. The results of the Hammett indicator method confirmed that the catalysts prepared using Na_2_CO_3_ had the highest basicity and, thus, the maximum number of active basic sites.

#### 3.2.2. Effect of Mg/Ca Ratio

The EDX data presented in Table 1 show that the Mg/Ca ratios obtained were in accordance with the actual values of Mg/Ca ratios taken in the feed while synthesizing the catalyst prepared using Na_2_CO_3_ and NaOH + Na_2_CO_3_ as precipitants. Meanwhile, the results obtained for the catalysts prepared using NaOH as a precipitant show that the Mg/Ca molar ratio of the catalysts was higher than that of the feed, which was caused by the greater affinity of Mg^2+^ ions for precipitation than that of Ca^2+^ ions [25]. This behavior could be supported by the XRD, FTIR, and TGA results showing the presence of MgO as the dominant compound, with CaO present only in trace amounts. The PC yield increased upon increasing the ratio from 0.5 to 1, and a further increase in the magnesium content led to a decrease in the PC yield. The increased catalytic activity might have been due to the increase in the specific surface area [25]. However, a decrease in catalytic activity at a Mg/Ca ratio of 2 was observed due to changes in the porosity and electronic properties of the catalyst, which eventually led to a decrease in its basicity [4].

#### 3.2.3. Effect of Basicity on PC Yield

The effect of basicity on the PC yield is displayed in Figure 4d. It was observed that the PC yield was high for the catalysts with high basicity. Thus, a relationship was established between the PC yield and the basicity. MgCa1CN was found to have the highest basicity, proving that basicity has a direct effect on the catalytic activity and, thus, on the yield of PC [38].

### 3.3. Effects of Reaction Conditions on PC Synthesis

The reaction temperature for PC synthesis varied from 150 °C to 170 °C, keeping the molar ratio of (PG)/(urea) at 4:1 and the catalyst dosage at 1 wt.% of (urea + PG) combined wt. The results in Figure 5a show that the PC yield increased with the increase in the reaction temperature from 150 °C, achieved a maximum of 95% at 160 °C, and then decreased upon a further rise in temperature to 170 °C. The reaction of PG and urea to produce PC is endothermic, so an elevation in temperature is favorable for the formation of products, as it will shift the reaction equilibrium forward. However, PG will vaporize faster at high temperatures, especially in a vacuum (PG will evaporate at a temperature lower than its boiling point (i.e., 188 °C) in a vacuum). This rapid vaporization of PG carries urea particles along and reduces the PC yield [39]. Therefore, the most suitable reaction temperature was found to be 160 °C.

The molar ratio of PG to urea varied from 1 to 5, and its effect on the PC yield is displayed in Figure 5b. The PC yield continuously increased with an increase in the molar ratio up to 4. However, upon further increase, the PC yield dropped to 86% at the ratio of 5:1. This can be explained by the fact that a very high PG/urea ratio will lead to an extremely low urea concentration, which will trigger side reactions and result in the formation of undesired products [40]. As a result, the reaction rate will decrease, resulting in a low PC yield. Therefore, the maximum PC yield of 95% was obtained at a 4:1 ratio for this reaction.

The effect of the catalyst dose on the yield of PC was studied by altering it from 0.5 wt.% to 3 wt.% (Figure 5c). The PC yield increased from 88% to 95% with an increase in the catalyst dosage from 0.5 wt.% to 2 wt.%. Upon further increasing the dose to 3%, PC yield decreased to 93% due to the rapid breakdown of urea into NH_3_ in the presence of the catalyst [26].

The reaction time for PC synthesis by urea alcoholysis over CaO–MgO was studied at regular intervals of time, keeping the rest of the parameters the same. Figure 5d displays the variation in the PC conversion and yield with reaction time. PC conversion was found to increase with time with an increasing PC yield. Increasing the reaction time beyond 4 h did not alter the PC conversion. Beyond 4 h, a decrease in the PC yield was observed due to polymerization of PC [10]. Thus, 4 h was selected as the optimal reaction time [39].

### 3.4. Regeneration and Reusability of the Catalysts

The stability of CaO–MgO was determined by consecutive reuse in the following manner: After completion of the first reaction of PC synthesis, the catalyst was recovered by centrifuge, washed with ethanol, dried in an oven at 60 °C for 6 h, and then reused. The catalyst was again reused by the same procedure two more times and then regenerated by calcining it at 700 °C for 5 h after drying according to the previously described procedure [40,41]. This regenerated catalyst was used for PC synthesis, and the recovered catalyst after completion of the reaction was reused again two more times, the results of which are shown in Figure 6. The PC yield was observed to decrease from 90% to 36% after the reuse experiment; nevertheless, the PC selectivity was found to be as high as 99%. No further experiments on catalyst reusability were performed, due to the marked increase in the catalysts’ density as a result of the species absorbed (e.g., polymerized PC) from the reaction medium.

Deformation of the structure can be one of the reasons for the decline in the activity of the spent catalyst. Some losses can also be incurred due to the physical mishandling of the catalyst. However, there was only a negligible change in the selectivity of the PC during the stability test. It can thus be concluded that changes in the structure of the spent catalyst did not change the reaction mechanism but controlled the extent to which the reaction was carried out.

The XRD results of the spent catalyst in Figure 2d show a decrease in the intensity of peaks corresponding to MgO and CaO. The decrease in the CaO peak was tremendous. A high-intensity peak for calcite at 2θ of 29.48° was observed. A possible reason for the presence of CaCO_3_ is that the CaO in the catalyst reacted with H_2_O to form Ca(OH)_2_, which further reacted with CO_2_ to form calcite [42].

## 4. Comparative Analysis

The performance of the CaO–MgO catalysts was compared with a few other catalysts reported to be used for PC production by urea alcoholysis. A summary of the critical parameters to provide comparative analysis of such catalysts is presented in Table 2. As can be seen, most of the reported catalysts (except Zn–Al oxide) require higher reaction temperatures for PC production as compared to our material. However, most of the listed catalysts containing Ma or Ca revealed significantly higher PC yields, including the CaO–MgO catalysts used in this study. It is also evident from the table that the best PG/urea ratio to achieve the maximum PC yield using varied catalysts is 4, which was also true in our case. Overall, it can be seen that the PC yield and selectivity observed in the present case were better than in the previously reported studies. Furthermore, high consistency was observed in the reactant ratio and catalyst mass required to achieve the maximum PC yield.

## 5. Conclusions

CaO–MgO catalysts synthesized using diverse precipitating agents (i.e., NaOH, Na_2_CO_3_, and a mixture of NaOH/Na_2_CO_3_) and different Mg/Ca ratios were tested for propylene carbonate production from urea alcoholysis with propylene glycol. The XRD results showed that the prepared catalysts exhibited highly crystalline structures consisting of cubic CaO and hexagonal MgO. The morphological characteristics of the diverse CaO–MgO catalysts as analyzed by FE-SEM revealed the existence of tiny particles of MgO and large aggregates of CaO. BET analysis showed that the catalysts prepared using equal molar ratios of Mg and Ca possessed the highest surface area of 31 m^2^/g. Batch reactor experiments conducted for PC production demonstrated an increase in the product yield as the PG/urea ratio was increased from 1 to 4. Similarly, a maximum PC yield of ~96% was achieved with a catalyst dose of 2 wt.% with respect to the reactor weight. It was also observed that the optimal time for achieving the maximum PC yield was 4 h, beyond which a reduction in PC yield was observed due to polymerization. Reusability studies on the best catalyst were also performed, which showed a reduction in the PC yield without any significant effect on PC selectivity. Overall, it can be inferred that the mixed oxides of Ca and Mg represent a promising candidate for the production of PC with high selectivity owing to their basicity, as investigated by the Hammett indicator method.

## Figures and Tables

**Figure 1 materials-16-00735-f001:**
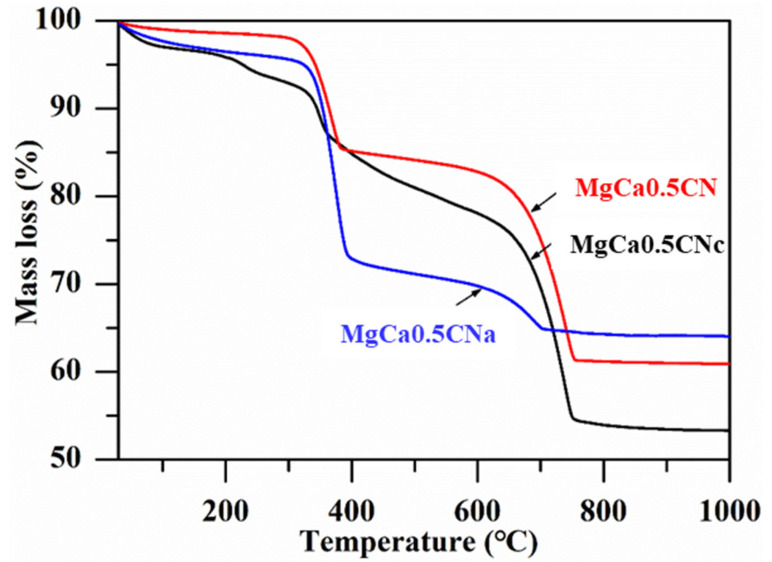
TG curves of the uncalcined catalysts.

**Figure 2 materials-16-00735-f002:**
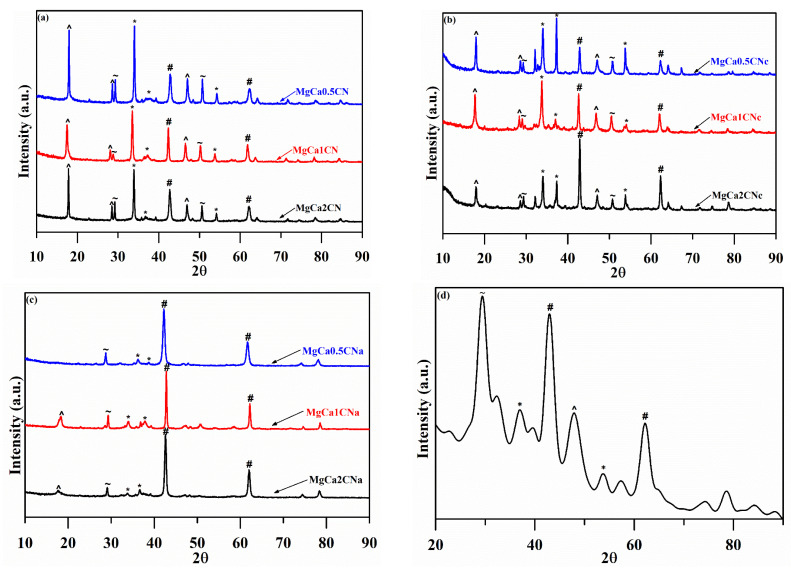
Powder X-ray diffraction patterns of CaO–MgO synthesized using (**a**) Na_2_CO_3_, (**b**) Na_2_CO_3_, and (**c**) NaOH as precipitants, calcined at 700 °C with different Mg/Ca ratios. (**d**) XRD patterns of the spent catalyst (^: Ca(OH)_2_, ~: CaCO_3_, #: MgO, and *: CaO).

**Figure 3 materials-16-00735-f003:**
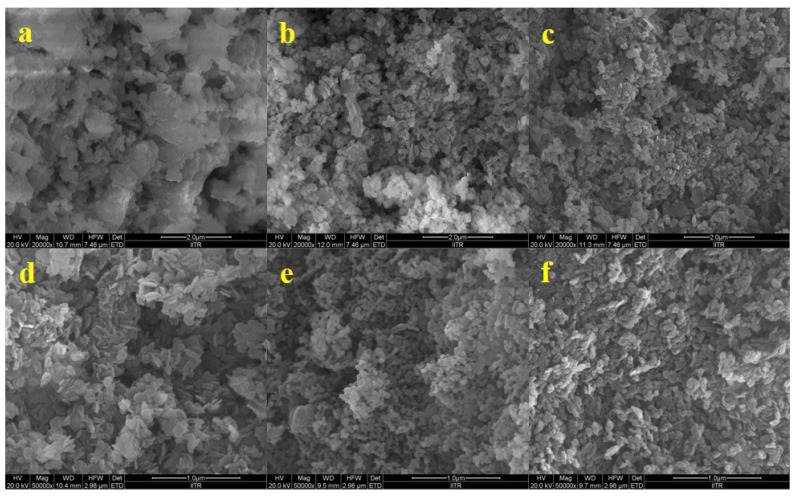
FE-SEM images of Cao–MgO catalysts before and after calcination ((**a**) MgCa_0.5_CNc, (**c**) MgCa_0.5_CN, and (**e**) Mg Ca_0.5_ before calcination; (**b**) MgCa_0.5_CNc, (**d**) MgCa_0.5_CN, and (**f**) MgCa_0.5_CNa after calcination).

**Figure 4 materials-16-00735-f004:**
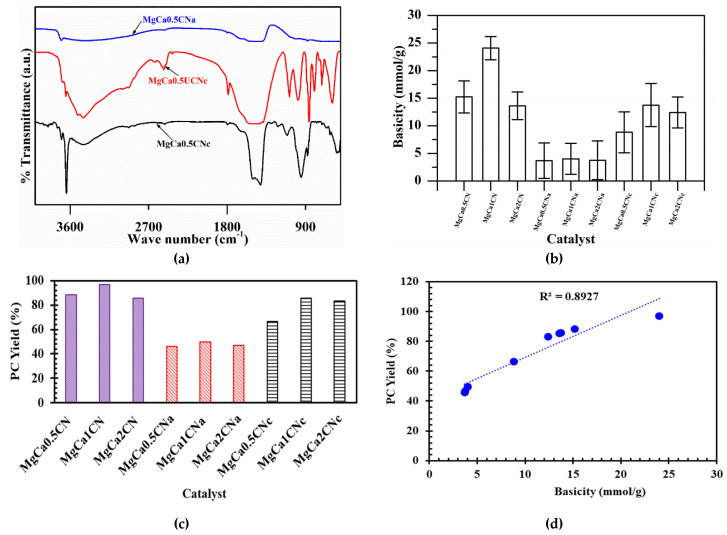
(**a**) FTIR analysis of calcined and uncalcined CaO-MgO. (**b**) Basicity of the catalysts. (**c**) Effects of Mg/Ca ratio and precipitating agents on the yield of PC; conditions: catalyst 1 wt.%, reaction time 3 h, PG/urea ratio 4:1, reaction temperature 160 °C. (**d**) Effect of basicity on PC yield.

**Figure 5 materials-16-00735-f005:**
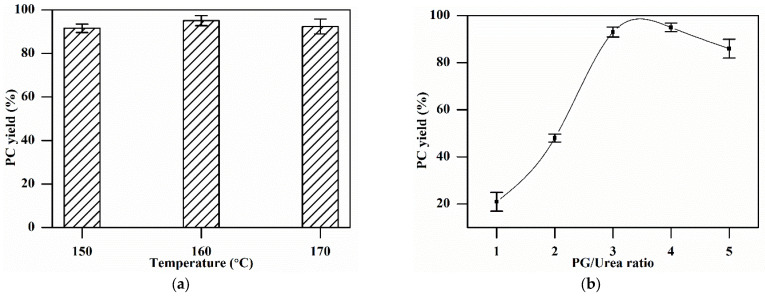
Effect of various parameters on the PC yield: (**a**) Reaction temperature: catalyst 1 wt.%, reaction time 3 h, PG/urea ratio 4:1. (**b**) PG/urea ratio: catalyst 1 wt.%, reaction time 3 h, reaction temperature 160 °C. (**c**) Catalyst dosage: reaction time 3 h, PG/urea ratio 3:1, reaction temperature 160 °C. (**d**) Reaction time: catalyst 2 wt.%, PG/urea ratio 3:1, reaction temperature 160 °C.

**Figure 6 materials-16-00735-f006:**
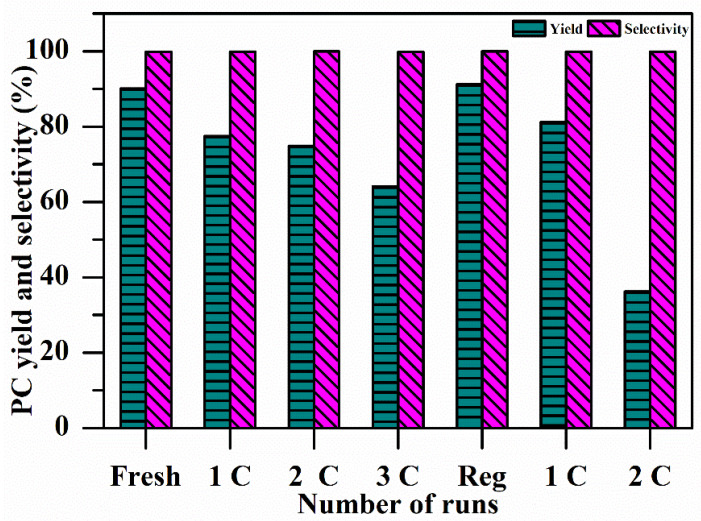
Effects of catalyst reusability on the yield and selectivity of PC. Conditions: catalyst 2 wt.%, reaction time 4 h, PG/urea ratio 3:1, reaction temperature 160 °C.

**Table 1 materials-16-00735-t001:** BET surface areas, EDX, and basic sites of the catalysts.

Catalyst	BET Surface Area (m^2^/g)	Basic Site (mmol/g)	Mg/Ca from EDX
MgCa2CNc	2	12.41 ± 2.8	1.3
MgCa1CNc	6	13.76 ± 3.9	0.83
MgCa0.5CNc	1	8.84 ± 3.7	0.476
MgCa2CN	25	13.62 ± 2.5	1.8
MgCa1CN	31	24.07 ± 2.1	0.96
MgCa0.5CN	29	15.23 ± 2.9	0.496
MgCa2CNa	26	3.76 ± 3.5	1.21
MgCa1CNa	31	4.01 ± 2.8	1.64
MgCa0.5CNa	26	3.68 ± 3.2	0.35

**Table 2 materials-16-00735-t002:** Comparative analysis for PC synthesis.

Catalyst	Reaction Time (h)	Temperature (℃)	Selectivity (%)	Yield (%)	Catalyst wt%	PG/Urea Ratio	Reference
Fluorinated MgAlO	3	160	95	91		4	[19]
Zn–Al oxide	5	140	96.8	87.4	3	4	[21]
Pb/Fe_3_O_4_/SiO_2_	2	180		87.7			[22]
CaTiO_3_–MgTiO= mixed catalyst	2	170	99	93.5	1.8	4	[27]
Pb–Zn mixed metal carbonate	5	180		96.3	1.8	4	[43]
Zinc chromium mixed oxide	4	170		97.8		1.5	[44]
Ca–Zn–Al oxide	2	170		82.9	2.7	2	[45]
La/HAP	2	170		91.5	1.5	4	[46]
CaO–MgO	3	160	99	96	2	4	This study

## Data Availability

The data will be made available upon reasonable request.

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
