# Peer review of "Efficient Propylene Carbonate Synthesis from Urea and Propylene Glycol over Calcium Oxide–Magnesium Oxide Catalysts"

_materials, 2023, doi:10.3390/ma16020735_

Round 1

Reviewer 1 Report

This work prepared propylene carbonate (PC) from urea and propylene glycol under the catalysis of Ca-Mg catalyst. Relevant number of experiments/characterizations have been done, and some significant conclusions have been drawn.

My main concern is the uncertainty of the results. The authors compared the effect of various preparing conditions, and some of them have shown little effect. I would suggest the authors do some uncertainty analysis of the results.

The reusability of the catalyst decreased rapidly in the runs. 3 and 2 runs are quite small number for evaluating the stability of the catalyst. It could be interesting to perform more runs to check the stability of the catalyst.

Author Response

  • Reviewer’s comment 1: My main concern is the uncertainty of the results. The authors compared the effect of various preparing conditions, and some of them have shown little effect. I would suggest the authors do some uncertainty analysis of the results.

Our statement for the mentioned reviewer’s comment 1:

Thank you for the comment. The author provided new plots with error bars in the revised manuscript to confirm uncertainty analysis in the experimental results.

  • Reviewer’s comment 2: The reusability of the catalyst decreased rapidly in the runs. 3 and 2 runs are quite small number for evaluating the stability of the catalyst. It could be interesting to perform more runs to check the stability of the catalyst.

Our statement for the mentioned reviewer’s comment 2:

The density of the catalyst (also mass loss) after 3 and 2 runs of reusability experiments was increased significantly due to the absorption of chemical species (e.g., polymerized PC) from the reaction medium. Hence no more experiments were performed. A similar statement has been provided in the revised manuscript in line no. 322-324 on page no. 11.

Reviewer 2 Report

Manuscript Number: materials-2058675

Full Title: Efficient propylene carbonate synthesis from urea and propylene glycol over calcium oxide-magnesium oxide catalyst

Remarks to the Authors:The authors studied the propylene carbonate (PC) synthesis via propylene glycol (PG) urea alcoholysis over CaO-MgO catalysts in an autoclave reactor. The highest achieved selectivity of PC was 99.8% and the conversion of PG was 96% with Mg/Ca = 1 at 160 °C, t = 4 h, catalyst amount = 2 wt.%, and PG/urea = 4:1. This manuscript is interesting and in the scope of the journal but at the same time has some weaknesses and ambiguities. In my opinion, this work can be considered for publication in MDPI Materials but only after revision. Please find my comments below.

1.The novelty of the manuscript is not completely clear. The authors should emphasize/disclose the novelty of their study in the Abstract and Conclusions sections.

2.The introduction section should include fresh references for the last 3 years.

3.What is the experimental error of the catalytic activity study (selectivity, conversion, yield)? Did the authors check the reproducibility of the obtained experimental data? The PC yield can not be so precise, e.g. 88.45%. Please correct it for the whole manuscript body.

4.What is the experimental error in the determination of the catalysts’ basicity based on the Hammett indicator method?

5.The authors used reference [29] for the determination of basicity but this reference related to acidity “Acidity of catalyst surfaces. Amine titration using Hammett indicators”. Please present the corresponding/correct reference and/or explain in more detail the used method for the evaluation of the basicity of catalyst samples.

6.The authors said: “The catalyst prepared from NaOH, Na2CO3, and a mixture of NaOH and Na2CO3 were named as MgCaXCNa, MgCaXCN, and MgCaXCNc, respectively with X denoting the Mg/Ca molar ratio”. Instead, I can observe in Figure 1 different names (e.g. MgCa0.5UCN, etc.) of utilized catalysts. Please, explain or correct it.

7.All labels on XRD patterns in Fig. 2 must be decrypted.

8.Table 1 must be corrected. The number of digits after the dot for the BET surface must be reduced. It is better to present only significant digits. What is the experimental error for BET surface area measurements? Also, I think the EDX method is not reliable for the quantification of the Mg/Ca ratio. The experimental error of this analysis can be tremendous.

9.Why do SEM images in Fig. 3 have different magnifications? It must be the same number if the authors decided to compare the catalyst samples with each other. By the way, the quality of SEM images including conditions (bottom part of images) should be improved.

10.The authors should compare their achieved catalytic activity results (conversion, selectivity, yield, catalyst weight, reaction time, temperature, etc.) with well-known literature data (recently published) and present it in a table format.

Author Response

  • Reviewer’s comment 1: The novelty of the manuscript is not completely clear. The authors should emphasize/disclose the novelty of their study in the Abstract and Conclusions sections.

Our statement for the mentioned reviewer’s comment 1:

Thank you for the valuable comments. The abstract and conclusion of the manuscript have been revised with emphasis on the novelty of the work reported.

  • Reviewer’s comment 2: The introduction section should include fresh references for the last 3 years.

Our statement for the mentioned reviewer’s comment 2:

Recent references have been added in the manuscript and at several instances in the manuscript.

  • Reviewer’s comment 3: What is the experimental error of the catalytic activity study (selectivity, conversion, yield)? Did the authors check the reproducibility of the obtained experimental data? The PC yield cannot be so precise, e.g. 88.45%. Please correct it for the whole manuscript body.

Our statement for the mentioned reviewer’s comment 3:

The authors have inserted new plots with error bars in the revised manuscript.

The PC yield, selectivity, and urea conversion rate values have been rounded off to the nearest significant numbers for the better representation of results.

  • Reviewer’s comment 4: What is the experimental error in the determination of the catalysts’ basicity based on the Hammett indicator method?

Our statement for the mentioned reviewer’s comment 4:

The author did not calculate the experimental error in the basicity analysis as had not thought of this at the time of the experiment. However, author will take care of such critical analysis in future publications.

  • Reviewer’s comment 5: The authors used reference [29] for the determination of basicity but this reference related to acidity “Acidity of catalyst surfaces. Amine titration using Hammett indicators”. Please present the corresponding/correct reference and/or explain in more detail the used method for the evaluation of the basicity of catalyst samples.

Our statement for the mentioned reviewer’s comment 5:

An appropriate reference has been cited in the revise manuscript.

  • Reviewer’s comment 6: The authors said: “The catalyst prepared from NaOH, Na2CO3, and a mixture of NaOH and Na2CO3 were named as MgCaXCNa, MgCaXCN, and MgCaXCNc, respectively with X denoting the Mg/Ca molar ratio”. Instead, I can observe in Figure 1 different names (e.g. MgCa0.5UCN, etc.) of utilized catalysts. Please, explain or correct it.

Our statement for the mentioned reviewer’s comment 6:

The names of the catalysts in Figure 1 has been corrected as suggested.

  • Reviewer’s comment 7: All labels on XRD patterns in Fig. 2 must be decrypted.

Our statement for the mentioned reviewer’s comment 7:

All labels in the XRD patterns are decrypted as suggested by the reviewer.

  • Reviewer’s comment 8: Table 1 must be corrected. The number of digits after the dot for the BET surface must be reduced. It is better to present only significant digits. What is the experimental error for BET surface area measurements? Also, I think the EDX method is not reliable for the quantification of the Mg/Ca ratio. The experimental error of this analysis can be tremendous.

Our statement for the mentioned reviewer’s comment 8:

The numerical values relevant to BET analysis has been rounded off to the nearest significant number as suggested. 

The authors have corrected the EDX data in the revised manuscript.

  • Reviewer’s comment 9: Why do SEM images in Fig. 3 have different magnifications? It must be the same number if the authors decided to compare the catalyst samples with each other. By the way, the quality of SEM images including conditions (bottom part of images) should be improved.

Our statement for the mentioned reviewer’s comment 7:

Authors have provided better quality SEM images with same magnification levels in the revised manuscript.

  • Reviewer’s comment 10: The authors should compare their achieved catalytic activity results (conversion, selectivity, yield, catalyst weight, reaction time, temperature, etc.) with well-known literature data (recently published) and present it in a table format.

Our statement for the mentioned reviewer’s comment 7:

A comparison table (Table 2) has been added in the revised manuscript in page no. 11.

Round 2

Reviewer 1 Report

The authors addressed well my comments and questions, I think it is eligible for acceptance.

Author Response

Thank you 

Reviewer 2 Report

The authors significantly improved their manuscript but still, I have the same question â„–5 without a proper response.

The authors utilized the reference [25] for the determination of basicity but this reference related to acidity “Acidity of catalyst surfaces. Amine titration using Hammett indicators”. Please present the corresponding/correct reference and/or explain in more detail the used method for the evaluation of the basicity of catalyst samples. There is no information in the proposed manuscript about the determination of catalyst basicity.

How many times did the authors repeat their titration? It should be at least 3 times. In this case, the authors can easy to provide/present the experimental error of the basicity calculation.

Author Response

As per the reviewer's comment, the results of Hammett indicators analysis are discussed on page no. 8 and relevant data is presented in Table 1 and figure 4b.

A suitable citation for the Hammett indicator analysis has been added.

Thank you for your suggestion.